# Pathophysiology of Peripheral Arterial Disease (PAD): A Review on Oxidative Disorders

**DOI:** 10.3390/ijms21124393

**Published:** 2020-06-20

**Authors:** Salvatore Santo Signorelli, Elisa Marino, Salvatore Scuto, Domenico Di Raimondo

**Affiliations:** 1Department of Clinical and Experimental Medicine, University of Catania, 95125 Catania, Italy; marinoelisa@msn.com (E.M.); salvatore.scuto1982@hotmail.it (S.S.); 2Division of Internal Medicine and Stroke Care, Department of Promoting Health, Maternal-Infant. Excellence and Internal and Specialized Medicine (Promise) G. D’Alessandro, University of Palermo, 90127 Palermo, Italy; domenico.diraimondo@unipa.it

**Keywords:** peripheral arterial disease, physical exercise, oxidative stress, heme oxygenase, antioxidants, pathophysiology

## Abstract

Peripheral arterial disease (PAD) is an atherosclerotic disease that affects a wide range of the world’s population, reaching up to 200 million individuals worldwide. PAD particularly affects elderly individuals (>65 years old). PAD is often underdiagnosed or underestimated, although specificity in diagnosis is shown by an ankle/brachial approach, and the high cardiovascular event risk that affected the PAD patients. A number of pathophysiologic pathways operate in chronic arterial ischemia of lower limbs, giving the possibility to improve therapeutic strategies and the outcome of patients. This review aims to provide a well detailed description of such fundamental issues as physical exercise, biochemistry of physical exercise, skeletal muscle in PAD, heme oxygenase 1 (HO-1) in PAD, and antioxidants in PAD. These issues are closely related to the oxidative stress in PAD. We want to draw attention to the pathophysiologic pathways that are considered to be beneficial in order to achieve more effective options to treat PAD patients.

## 1. Methodology of Literature Search for Review

### 1.1. Data Sources and Search

In order to tackle the above-described issue, a thorough literature search strategy has been laid out by a team which has considerable experience in the analysis and research of medical papers, especially in the consultation of the medical scientific web platform (MEDLINE). Such research included recent published papers or reviews dating up to 2019, using a combination of key words (e.g., peripheral arterial disease, inflammation, biomarkers, pathophysiology, and therapy). The search was limited to papers published in English. 

### 1.2. Data Extraction

Each single participant in the literature search extracted the most pertinent content whilst others verified the accuracy and completeness of the extracted data. Each author analyzed whether the search results were different or confounding in order to release a complete overview of the field. Such a peer reviewed strategy helped to identify and extract the data that could be deemed as most meaningful for the research.

## 2. Introduction on Topic

Peripheral arterial disease (PAD) is one of the clinical types of atherosclerotic diseases. For this reason, particular attention should be given to its frequent diagnosis in elderly individuals, with particular prevalence of PAD-affected patients in socially and economically advanced countries [1,2].

PAD is often under-diagnosed, although we are in possession of non-invasive diagnostic techniques such ultrasound examination by measuring the ankle brachial index (ABI), which is an easy and repeatable tool helpful in diagnosing PAD as well as in monitoring the outcome of PAD patients [3]. Scientific evidence shows that a considerable number of individuals are not aware of having or suffering from the symptoms that could be associated to PAD [4].

It is important to highlight the close link between PAD and a high risk of acute cardiovascular events, as shown by the frequency of coronary and carotid ischemic events occurring in PAD patients [5].

Guidelines on PAD treatment suggested the use of many drugs (statins, aspirin, clopidrogel, dual anti platelet drug therapy, cilastazol, pentoxyfilline, nifedipine); however, efficacy in the improvement of the symptoms (intermittent claudication, pain free walking distance) or long term outcome (cardiovascular risk, cardiovascular acute event) is still being debated. 

PAD patients benefit from regular supervised physical exercise (PE) as an effective option to improve the muscle performance, to reduce the free pain walking distance, and to counteract the intermittent claudication. Therefore, PE ameliorates the quality of life. PE plays a crucial role in the cure of the PAD, being both a preventive as well as a mitigating factor. To clarify the positive effect originated by the PE in PAD we must to draw our attention to the presence of the oxidative stress (OxS) as a key mechanism having a role in PAD pathophysiology. It is known that PAD pathophysiology has shifted from a hemodynamic scenario towards endorsing OxS [6]. In this review, the authors aim to provide some scientific thoughts on the PE biochemistry, on pro and anti oxidative effects from the PE, and on the OxS and PE in patients with the PAD. Moreover, the authors will debate antioxidant treatments in PAD, and heme oxygenase 1 (HO-1) in PAD.

## 3. Biochemistry of the Physical Exercise: Pro and Anti Oxidative Effects

The close relationship between muscle stress and arterial wall damage and cardiovascular events has been scientifically demonstrated [7]. PE acts on arterial resistance inducing vasodilating effects, modulates the arterial pressure, improves both the insulin resistance and fat metabolism, and acts on the adipose system. PE positively regulates systemic low-grade inflammation, reduces the high circulating levels of pro-inflammatory cytokines, counteracts the endothelial dysfunction, reduces platelet adhesion and aggregation, and improves sympatho-vagal balance [8]. PE regulates the arterial pressure, reduces the high plasma level of lipid and lipoproteins, and reduces both overweight and obesity. PE gives its notable positive impact on the so-called risk factors for cardiovascular diseases (CVDs). PE counteracts the physical inactivity that leads to clinical conditions through dysregulation of such molecular ways.

PE shows anti oxidative activities such as modulation of the antioxidant enzymes (mitochondrial superoxide dismutase Mn-SOD, Cu/Zn-SOD, catalase, glutathione peroxidase) [8,9].

PE improves the activation of the nucleotide adenine dinucleotide phosphate oxidase [(NAD(P)H] oxidase) by enhancing its antioxidant capability [10]. PE acts on endothelial functions by regulating endothelial genes which are effective in modulating oxidative metabolism, cell apoptosis, cell growth and proliferation, and endothelial vascular nitric oxide synthase (eNOS) [11,12,13,14,15,16,17,18].

PE also spreads its effects on arterial wall remodeling, inducing angiogenesis and arteriogenesis [19]. 

Concerning the positive activities of the PE, it was recently proposed that the protective effects of PE could also be attributed to the muscular release of the peptides called “myokines”. These molecules, secreted during skeletal muscle contraction, may trigger specific metabolic pathways in different tissues and organs far from the muscle allowing to communicate with many organs such as visceral fat, bone, liver, and nervous system, among others [20,21].

Based on current knowledge, there is growing evidence of myokines in humans, and more so on the biological role of interleukin-6 (IL-6) and the broad range of metabolic and anti-inflammatory effects. Almost all effects were demonstrated to be related to acute exercise, whereas there is low evidence regarding effects due to regular training in decreasing plasma levels of IL-6 [22].

Myokines work as an endocrine system. IL-6 was firstly identified as a muscle derived myokine, and released into the bloodstream during muscle contraction. Muscle-derived IL-6 blood concentration results are directly proportional to the intensity of the exercise, and also depend on the type of exercise. Muscle-derived IL-6 shows great pro-inflammatory activities, such as that demonstrated in sepsis (it represents a key biomarker of systemic inflammation associated to unfavorable metabolic effects). Muscle-derived IL-6 favors the release of anti-inflammatory cytokines: IL1 receptor antagonist (IL1-ra) and IL-10. The blood release of these agents was initially considered as a simple exercise-induced muscle damage; however, it should be considered as mainly metabolic support of the muscular metabolism during exercise, favoring glucose availability, lipolysis and oxidation of fat [23].

Focus on the role played by the IL-6 as relevant myokine, provides an adequate outline to understanding myokines. Interestingly, PE achieves two important objectives concerning the glucose metabolism. PE improves insulin sensibility and body-weight control due to the favorable metabolic profile induced by muscle contraction both during and after exercise [24].

Regular and moderate PE raises both adenosine triphosphate (ATP) activity and oxygen extraction from tissue. Thus, PE can play a positive role in managing the CVDs [25]. 

Maximum and repeated muscle exercise negatively effects antioxidant agents; it raises reactive oxygen species (ROS) generation and thus the unregulated or strenuous PE is strongly linked to the OxS [26]. Therefore, higher levels of both the ROS and glutathione oxidation resulting from strenuous physical exercise mark the activated pro-oxidative status [27]. 

Conversely, the regular moderate PE upregulates antioxidant genes, and promotes adaptive mechanisms. This positive effect is clearly demonstrated by the expression of genes which code for antioxidant enzymes (i.e., superoxide dismutase, catalase, glutathione peroxidase), of the adaptive molecules (endothelial nitric oxide synthase, inducible nitric oxide synthase) [28]. Regular and moderate PE must be considered as an anti-oxidant player, as shown by the role played by ROS in cell signaling, in gene expression regulation, and by the favored cell adaptive capability [28].

Positive epigenetic effects, systemic adaptive response, increased antioxidant capability, and improved resistance to OxS cumulatively favor the positive health effects originating from PE [29].

## 4. Oxidative Stress and Physical Exercise in Patients with Peripheral Arterial Disease

Arterial stenosis caused by atherosclerotic plaque build-up in peripheral arteries is crucial in determining the hemodynamic disturbance of the arterial flow of peripheral circulation. Hemodynamic peripheral disturbance is paramount in provoking severe damage to skeletal muscle in patients suffering from PAD [30,31,32,33,34,35]. Moreover, PAD patients demonstrated either low or high grades of inflammation, and active OxS [36,37,38,39,40], which are two pathophysiological mechanisms in PAD. The hemodynamic disturbance of peripheral circulation in PAD characterizes the chronic ischemia, which in turn damages the myofibers of the lower limb skeletal muscles. Differences in hematic loads varying with muscle tissue need (ischemia) is crucial in causing intermittent claudication (walking pain) which is the major clinical symptom in PAD patients. Repeated episodes of ischemia lead to progressive and severe damage to skeletal musculature and the dysfunction of skeletal muscle cell mitochondria [41]. Findings from the muscle biopsies of PAD patients showed great muscle-cell apoptosis and reduced type-I myofiber, both of which may interfere with muscle performance [42,43]. Mitochondrial dysfunction of skeletal muscle cells contributes to impaired muscle metabolism in PAD. High circulating and muscle levels of the intermediates of oxidative phosphorylation, including acyl carnitines, found in PAD patients, suggested lowered mitochondrial metabolism [44]. The mitochondrial mass of skeletal muscle in PAD is higher and, by contrast, there is lower activity of mitochondrial complexes impeding ATP generation, and enhancing ROS generation. Altered mitochondrial function restricts oxygen utilization and it may bring endothelial dysfunction because mitochondrial-derived oxidants reduce nitric-oxide bioactivity [45,46]. Muscle myofibers degeneration is associated with the OxS generation, including carbonyl groups, 4-hydroxy-2-nonenal adducts and protein modifications produced by ROS [47]. Active mitochondrial capability is important in angiogenesis because it is consistent with the notion of coupling vascular and muscular parameters. In hind limb ischemia models, peroxisome proliferator-activated receptor gamma coactivator 1-alpha (PGC-1a), a key regulator of mitochondrial biogenesis, promotes vascular regeneration [48]. Skeletal muscle dysfunction, including mitochondrial abnormalities, affects walking ability in PAD. Both decreased calf muscle content and altered fiber type relate to reduced functional parameters. Importantly, mitochondrial dysfunction assessed by magnetic resonance spectroscopy to evaluate phosphocreatine recovery is associated with lower treadmill walking times. PAD patients with greater amounts of muscle acyl carnitine accumulation have greater degrees of exercise limitation. Evidence of myofiber damage is associated with both reduced walking distance and muscle strength in patients with claudication. Furthermore, the altered regulation of a cytoskeletal protein, desmin, is associated with reduced mitochondrial respiratory function and functional capacity in PADs [49]. There is evidence of inadequate mitochondrial clearance through autophagy in skeletal muscles of PAD patients that in turn is associated with the walking performance, and it is also consistent with increased mitochondrial damage (Figure 1). Higher levels of daily activity are associated with healthy calf muscle parameters. Several aspects of skeletal muscle phenotype, including increased calf muscle fat and decreased muscle density, predicted a 2-year functional decline in a longitudinal study. Evidence of reduced mitochondrial biogenesis is associated with higher overall mortality, which is potentially mediated through reduced physical activity [42]. 

The efficacy of PE in managing PAD has been long debated. Severe damages to skeletal myofiber and raised ROS plasma levels were induced by maximum muscle exercise in PAD patients [43,44]. Researchers have demonstrated the strong relation between sedentary no PE with lipid peroxidation and superoxide enzyme generation [45]. Both elements could be seriously dangerous for the arterial wall, and they could lead to CVDs. [46]. Conversely, PE origins the increased expression of enzymes such as superoxide dismutase (SOD), catalase (CAT) and gluthatione peroxidase (GPx) [47]. Method, intensity, and regularity of the PE have diverse effects on oxidative balance. The acute bout of vigorous exercise is strong pro-oxidant mechanism leading to the massive and fast increase of the biomarkers of OxS [48]. It should be noted that high intensity and discontinuous physical training have less impact on the redox system than continuous moderate-intensity physical training [49]. 

Thus, the acute PE *di-per-se* acts as pro oxidative agent leading to dangerous biochemical dysfunctions [50]. 

Regular PE induces a progressive and stable adaptive situation. As a consequence, regular supervised PE upregulates the total antioxidant capability. Enhanced anti-oxidant capability originating from regular PE progressively buffers the pro oxidative unbalance that characterizes the chronic ischemia of muscle tissue originating from the chronic reduction of the arterial perfusion of arteries of the lower limbs [51]. The physiologic adaptive conditions originating from regular PE are mainly epigenetic; thus, such genetic transcription pathways seem to be involved in adaptation to the exercise. Several factors such as the nuclear kB factor (NF-κB), the mitogen activated protein kinase signaling pathways (MAPK) to upregulate catalase enzyme, Mn-SOD, GPx, glutathione antioxidant enzymatic complex, as well as the inducible nitric oxide synthase were activated by the PE [52].

There are contrasting opinions about the effects provoked by the PE on white cell viability. Acute stress by PE could induce the apoptosis of lymphocyte by damaging mithocondria through oxidative mediated pathway [53]. Intensity of the PE has different effects on white blood cells and inflammation. In fact, the extreme PE raises such biomarkers as the mieloperoxidase marker of the white blood cell degranulation, C reactive protein as a marker of the acute phase of infllammation, and the pentraxin 3 known inflammatory biomarkers. In contrast, moderate or attenuate physical training showed the low level of the above parameters [54]. 

Concerning the effects originated by strenuous or maximal PE on inflammatory biomarkers, it could be interesting to analyze the results from study designed to evaluate on effects of the maximal treadmill test conducted until muscle discomfort occurrence both in PAD patients and in healthy individuals. The results of this study demonstrated higher levels of IL-6 and tumor necrosis in PAD compared to control subjects. Different biomarkers were found to be raised when the pain (in PAD patients) or maximal discomfort (controls) occurred in the lower limbs. The results confirmed that activation of the white blood cell occurred in acute stressed circulation of peripheral arteries [55]. Inflammatory activation seems to be correlated to different muscle capability, as demonstrated by moderate inflammatory response measured after treadmill test in healthy individuals, and in individuals showing no severe intermittent claudication. On the other hand, response increased in severe claudicants. 

Measurement of the biomarkers could play an interesting role as a useful marker in grading the chronic ischemia [56]. Furthermore, the effect of the PE on oxygen tissue extraction has been demonstrated, and it is very interesting to note that individuals having limited muscle performance (such as PAD patients) achieve the maximal tissue oxygen extraction measured in calf muscles after the treadmill test more quickly, whilst there is a delayed recovery time for the oxygen extraction [57].

There is evidence concerning the positive effects of regular supervised physical training on physical performance, on clinical outcome, and on the long-term prognosis of patients affected by CVDs including PAD [58]. It is notable that the positive effects initiated by supervised PE in PAD were estimated by lowered ROS generation, and reduced levels of inflammatory markers [59]. There is evidence on positive activities originating from PE on clinical targets of the PAD, such as improvement in walking distance (pain the pain) reducing the patient’s discomfort, and in the skeletal muscle performance. Finally, the PE ameliorates the quality of life of the PAD patients.

Based on evidence, regulated supervised physical training is now listed as a class IA option in treating PAD patients [60,61].

## 5. Antioxidants, and Heme Oxygenase 1 in Peripheral Artery Disease

PAD patients suffer from modified acetyl-CoA ester accumulation when the concentration of carnitine in muscle cells lowers [62]. In PAD patients there is inadequate ATP generation. Thus, cell respiratory activity worsened. PAD patients show increased levels of esterified derivatives of acyl-CoA, which may be closely related to lowered blood perfusion [63].

This metabolic imbalance occurs when muscle and plasma levels of carnitine are low, as, e.g., in patients suffering from progressed PAD [64]. The results of studies suggest that carnitine stimulates glucose disposal and oxidation, leading to the efficient utilization of glucose under ischemia as occurs in PAD patients. The anti-oxidative drug propionyl L-carnitine was shown to modify OxS in PADs [64,65,66].

It is noteworthy to clarify the role played by biochemical agents in cardiovascular tissue. It has been demonstrated that OxS characterized PAD as the higher levels of the nitric oxide 2 enzyme (NOX2) found in PAD patients compared to normal subjects [67]. In PAD, there is an upregulation of the NO bioavailability, and thus an improvement of the NO synthesis was the target to achieve a treatment of the PAD patients.

Different antioxidant agents and drugs were tested in studies forwarded to evaluate the OxS inhibition. Vitamins C [68,69,70] and E [71,72], glutathione [73,74], natural agents as the polyphenols (epicathechin, catechin) [75,76], and carnitine were tested to counteract the OxS both in clinical trials or in the HUVEC model. Antioxidant agents and drugs showed several anti OxS effects concerning the clinical performances (walking distance, pain free distance) linked to the OxS, on endothelial dysfunction (microculatory perfusion, flow mediated dilatation, arterial response to exercise, platelet dysfunction/aggregation), and on surrogate oxidative biomarkers bloodstream released, i.e., malondhyaldheide, 4-hydroxynonale, and TABRS (Table 1). The supplementation of anti OxS agents could be evaluated as an additional and helpful option in threating PAD patients. 

We would like to draw attention to the HO-1 protein, since PAD patients show low HO-1 plasma levels. This seems to match with the differences found in lactic acid plasma levels in PADs and non-PADs. In relation to the OxS markers, we would like to highlight glutathione (GSH) levels in PADs. We found lowered GSH and higher plasma levels in progressed PAD patients (at the 2nd B of Leriche’s classification) than in PADs at the 2nd A stage [77]. We postulate that the reduced HO-1 levels may reflect the reduced intracellular content in PADs. Moreover, severe metabolic tissue disorders such as OxS initiated by chronic repetitive (intermittent claudication, pain occurrence walking related) ischemia is a characteristic of PAD patients. HO-1 is known as a rate-limiting enzyme for heme degradation [78,79,80]. Numerous accumulated evidence has demonstrated its protective capabilities for the cardiovascular system. Since patients having CVDs demonstrated a low level of bilirubin, it was postulated that anti-oxidant activity of HO-1 can be due to biliverdin, bilirubin, and CO [81]. However, results from metanalysis including eleven studies showed no causal relationship between bilirubin level and CVDs. Meta-analysis concluded that exhausted anti-oxidant proprieties are causative factors to increase the ROS generation occurred in CVDs [82].

Overexpression of HO-1 achieves both attenuation of the increase of the inflammatory mediator generation, and improvement of vasodilating response to oxidative agents such as the oxidized low density lipoproteins [83]. HO-1 antagonizes the remodeling of the arterial wall and dysfunction of endothelium, and protects the vessel walls from pathological remodeling and endothelial cell dysfunction [84]. 

It is known that disturbance of laminar arterial flow plays a crucial role in promoting the adhesion of various blood cells to the arterial wall, and plays a role in arterial plaque generation and for its growth or vulnerability. The anti-oxidant protective capabilities of the HO-1 can be helpful in peripheral disturbed arterial circulation, as occurred in PAD (Figure 2).

Since low levels of HO-1 were found in PAD patients, these data could promote more actions concerning HO-1 as therapeutic option. HO-1 inhibits the lipid peroxidation, as demonstrated by the lowered receptors for low-density cholesterol in the mice knockout model [85]. This was also demonstrated by the improvement in nitric oxide synthesis under hyperchelosterolemia [86] recorded after exposure to oxidized low density cholesterol, and by pro-oxidant transition metal [87] activity, which are the most significant mechanisms activated by this enzyme in cardiovascular protection. There is interest in inducible forms of HO-1 as an anti-inflammatory protein sensitive to OxS induced by various agents on the cardiovascular system [88]. The lack of effects from Vitamins E and C on outcome of patients with CVDs, and on suppression of the ROS generation, draws attention to HO-1 as an intrinsic defense system in artery walls. HO-1 combats the progression of atherosclerotic diseases since anti-inflammatory, anti-oxidant, anti-apoptotic, and anti-thrombotic properties have been demonstrated [89,90,91,92,93]. Low levels of the HO-1 are expressed in most tissues under basal conditions; however, this enzyme is highly inducible in response to various pathophysiological stimuli. The degradation of the pro-oxidant heme generation of the antioxidants biliverdin and bilirubin and the production of vasodilator CO are crucial mechanisms in protecting against the progression of atherosclerosis. Data from animal models showed that a lack of HO-1 resulted in accelerating atherosclerosis. Redox-sensitive transcription factor known as nuclear factor erythroid 2-related factor (Nrf2) regulates HO-1. Nrf2 is kept in a latent state by interaction with Kelch ECH (Keap1), and its associated protein is a repressor protein. OxS stimuli lead to a change in Keap 1, resulting in Nrf2 release. Cytosolic Nrf2 is translocated into the cell nucleus where it binds to the antioxidant response element (ARE), thereby initiating the transcription of antioxidants including HO-1, superoxide dismutase (SOD), catalase, and NAD(P)H quinone dehydrogenase 1 (NQO1). HO-1 levels were inversely associated with PAD [93] and multivariate analysis showed how HO-1 was an independent predictor of the presence or severity of PAD [94]. It is still unclear how effective this mechanism is in inducing low HO-1 plasma levels in patients with PAD, but bearing in mind that PAD patients suffer from a chronic reduction in hematic load, these patients also experience repeated increased ischemic conditions initiated by muscle exercise (normal walking, walking test). The defensive capability of HO-1 in responding to OxS results in attenuating these ischemic situations. The chronic and long duration of the mitochondrial stress on skeletal muscle cells produces reduced performance of the HO-1 anti-oxidative defense system. Studies performed in animal models of limb ischemia have shown interesting results from gene and cell therapy with HO-1. Thus, HO-1 inducers could be considered in treating patients with PAD.

## 6. Discussion

OxS is involved both in triggering and developing the atherosclerosis. Consequently, any agents, drugs and strategies must counteract its deleterious effects on arterial functions. The efficacy of medical strategies in treating PAD is currently debated. The most important objectives of the following long-term outcomes of PAD patients are the reduction in CVD events, and the potency of open peripheral arterial circulation as result of open or interventional procedures. On the other hand, OxS produced by chronic peripheral ischemia induced by haemodynamic imbalance must be fought to restore cell and tissue metabolism. New medical strategies must be directed towards achieving objectives such as improving oxygen tissue extraction and creating angiogenesis or arteriogenesis. In this regard, the regular supervised PE has shown interesting effects in treating PAD patients: it improves physical and muscle performance and acts on cell and tissue metabolisms, such that regulated physical exercise seems to counteract OxS. There is great interest in the vaso-protective effects of HO-1, which is largely attributable to its end products, being a potent antioxidant and anti-inflammatory and also by affecting the proliferation, migration and adhesion of smooth vascular muscle cells, endothelial cells, and leukocytes [94,95,96,97,98]. It is known that the redox capability of HO-1 is regulated by Nrf2 [95]. The protective role of HO-1 in several atherosclerotic diseases was still released; however, a few data were still released concerning HO-1 in PAD. Poor attention to this field could be caused by low prevalence of PAD compared to other atherosclerotic diseases. Consequently, it seems there is less attention paid to PAD in searching and in diagnosing other atherosclerotic diseases (coronary, carotid diseases).

Since pharmacotherapy for PAD patients failed to fully achieve some objectives (walking distance improvement, pain reduction or absence, cardiac or carotid diseases morbidity, PAD’s outcome and prognosis, quality of life) there is an interest in new therapeutic options such as targeting the HO-1. In fact, a number of studies have clarified the interesting capability of the induced HO-1 through dietary antioxidants (i.e., curcumin, polyphenols, isothyocianates), through PE, and through some available drugs (statins, fibrates) [96,97,98,99,100,101,102,103,104,105]. We are looking for new advanced drugs derived from further study and encourage researches to undertake this work. 

## 7. Conclusions

Pathophysiology of PAD is complex as it includes hemodynamic disturbances such as reduced hematic load, progressed reductions of muscle tissue perfusion, damage of muscle fibers, and reduction of cell respiratory capability. Furthermore, distributive arterial circulation and nutritional circulation are progressively worsened and severely dysfunctionated. The progressed knowledge on oxidative biomarkers released in the bloodstream, and on inflammatory biomarkers in causing endothelial dysfunction suggest to us that screening the OxS both in symptomatic and in asymptomatic PAD patients will be a helpful tool to monitor the efficacy of treatments for PAD.

HO-1 was clearly evaluated as protective against atherosclerotic damage [97,100,101]; thus, it could be an interesting option to treat patients with PAD. 

The last concern could be closely related to the known positive effects originating from the supervised PE both on clinical performance and on hypoxic adaptation. 

We want to encourage more studies on OxS and on oxidative biomarkers; we hypothesize that results could help us to improve and enhance knowledge of the complex pathophysiology of PAD.

## Figures and Tables

**Figure 1 ijms-21-04393-f001:**
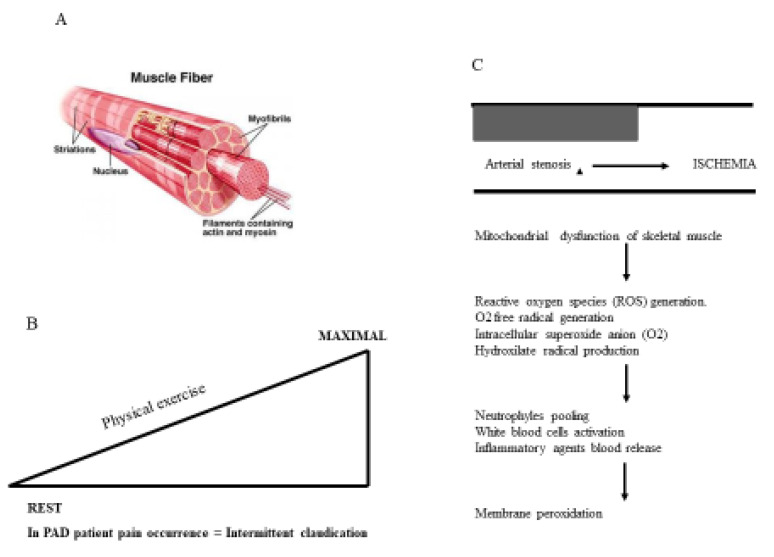
(**A**) Skeletal muscle contraction. (**B**) Effects of physical exercise (PE) in peripheral arterial disease (PAD). Maximal PE induces the intermittent claudication (i.e., characteristic symptom in PAD). (**C**) Mitochondrial, cell and membrane dysfunctions induced by arterial stenosis in a PAD patient.

**Figure 2 ijms-21-04393-f002:**
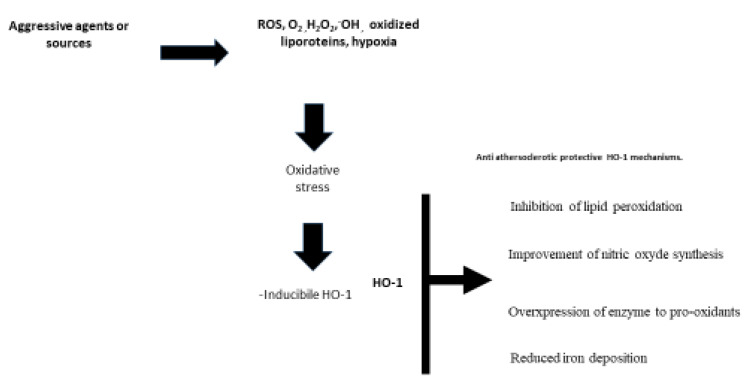
Figure shows the source and agents able to induce the oxidative stress. The picture briefly summarizes the capabilities of HO 1 in protecting against the atherosclerotic process. **ROS** = reactive oxidized species, **O_2_** di-oxygen (singlet), **H_2_O_2_** = peroxide, **OH** = hydroxide ion, **HO-1** = heme oxygenase 1.

**Table 1 ijms-21-04393-t001:** The below table summarizes data from studies focused on effect of antioxidant supplementation in PAD.

Antioxidants	Effects and Markers	References
**Propionyl-l-Carnitine**	FMD and brachial basal diameter significantly increased Increase in NOx bioavailability Decrease in 8-OHdG	[62,63,64,65,67]
**Vitamin C**	Reduces OxS walking induced Reduces arterial pressure response to physical exercise No reduction of flow mediated dilatation (FMD) by maximal physical exercise No elevation of TABRS OxS marker No elevation of soluble CMA-1	[68,69,70]
**Vitamin E**	Reduces OxS walking induced	[71,72]
**Gluthatione**	Reduces pain free walking distance Improves macrocirculatory flow after physical exercise	[73,74]
**Polyphenols:** **Epicatechin** **Catechin**	Enhances platelet activation Increases the release of soluble cell adhesion molecules (sCAMs) Decreases eNOS activation Effects on NO bioavailability	[75,76]

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
