# Peer review of "Pathophysiology of Peripheral Arterial Disease (PAD): A Review on Oxidative Disorders"

_ijms, 2020, doi:10.3390/ijms21124393_

Round 1
Reviewer 1 Report
This is an interesting topic for review. However, the manuscript is very difficult to read, comprehend, and evaluate, due to it being written in poor English.
Also, the review in some sections seem to be too superficial. For instance, "Physical exercise: biochemistry" section could have more details about genes and enzymes being affected by different excersie schedules/types (e.g. repetitive vs. non-repetitive, prolonged vs. short-term, aerobic vs. anaerobic etc).
Some controversies encountered in the same section of the manuscript could be discussed better. For instance, it is stated that "PE reduces the levels of IL6" (line 74, page 2) which is well-known to have a pro-inflammatory effects. Later, it is stated that "IL6 is the most commonly known of the myokines being released into the bloodstream by physical muscle exercise. Myokine IL6 favors the release of anti-inflammatory cytokines" (lines 83-8, page 2). It would help to incorporate some thoughts of authors reviewing the literature on these controversial matters.
Author Response
Subject: manuscript ID ijms-791017
Reviewer 1.
Q.1 the review in some section seems to be to superfial. For istance”Physical exercise biochemistry” section coul have more details about genes and enzymes being affected by different exercise schedules/types (e.g. repetetive vs non-repeteitive, prolonged vs short-term, aerobic vs anaerobic etc).
Reply. I thank you for question.
It is correct to discuss on effects on the biochemistry of skeletal muscles originated by different physical exercises. On this concern I briefly summarized effects originated ny physical exercise on enzymes, having anti oxidant activity, on enzymes released from endothelial menbrane. Additionally, I included into the section data from studies on effect of physical exercise on genes having capability on oxidation, or promoting cell growth or cell apoptosis , and playing a role to regulate activity of the endothelial enzymes I included into the review the section listed as “Physical exercise biochemistry” to summarize briefly most relevant knowldge on biochemical mechanisms related to physical exercise. The section must be used to introduce follow sections mainly focused on relationship between the physical exercise with osidative stress and with arterial disease (i.e. peripheral arterial diseaase).
Q.2 Some controversies encoutered in same section of the manuscript coul be discussed better. For istance, it is stated that “PE reduces the levels of IL 6 (line 74,page 2) which is well-known th have a pro-inflammatory effects. Later, it is stated that “Il 6 is most commonly known of the myokines being released in the bloodstream by physical muscle exercise. Myokine 6 favors the release of anti-infllamatory cytokines” (lines 83-88, page 2). It would help to incorporate some thoughts of authors reviewing the litterature on thse controversial matters.
Reply. I thank you for question.
I rewritted the section according to suggestions from review.
Kind regards.
Prof. Salvatore Santo Signorelli MD
Reviewer 2 Report
Figure 1 is very unclear and confusing. Please remake this figure.
Please provide more details in figure legends for both Figure 1 and 2, so that they can be understood alone.
Author Response
Subject: manuscript ID ijms-791017
Reviewer 2.
Q.1 Figure 1 is very unclear and confusing. Please remake this figure.
Reply. I thank you for suggestion to improve quality and understanding of figure.
Although I used similar scheme in remaking figure, I rephrased internal legend.
I hope that I new version of figure may to improve its undestanding.
Q.2 Please provide more details in figure legends for bth figure 1 and 2, so they can be understood alone.
Reply. I thank you for suggestion in providing details to improve understanding.
I rewritted legends and details for both figures
Kind regards.
Prof. Salvatore Santo Signorelli MD
Round 2
Reviewer 1 Report
Although the text was edited, more extensive editing (including punctuation check) is required, as it is still difficult to comprehend and evaluate.
Overall, the review is too superficial and could be improved by providing more details for each major statement/sub-topic within the sections.
Also, it requires adding more structure to it, so that the reader could follow the thread of the main message of the section, rather than simply listing out not well-connected facts, and jumping from one fact to another and back.
Besides, distribution of content between sections should be revised to have maintain certain structure and prevent mixing/overlapping. For instance, “3. Physical exercise: biochemistry” has intro and 2 sub-sections “3.1 PE and oxidative stress” and “3.2. Skeletal muscle in PAD”; followed by next section “4. Oxidative stress and PE in patients with PAD”. The intro of section 3 and 3.1 could be merged together as they both mainly list pro- and anti-oxidative effects of PE; whereas 3.2. could be sub-section of section 4 as they both discuss PAD and its effects on oxidative status in muscle.
Author Response
Manuscript. ijms-791017.